# EXPRESSIVE MONOTONIC NEURAL NETWORKS

**Niklas Nolte**[*]**, Ouail Kitouni**[*]**, Mike Williams**
The NSF AI Institute for Artificial Intelligence and Fundamental Interactions
Massachusetts Institute of Technology
Cambridge, MA 02139, USA
`{nnolte, kitouni,mwill}@mit.edu`

## ABSTRACT

The monotonic dependence of the outputs of a neural network on some of its inputs is a crucial inductive bias in many scenarios where domain knowledge dictates such behavior. This is especially important for interpretability and fairness considerations. In a broader context, scenarios in which monotonicity is important can be found in finance, medicine, physics, and other disciplines. It is thus desirable to build neural network architectures that implement this inductive bias provably. In this work, we propose a weight-constrained architecture with a single residual connection to achieve exact monotonic dependence in any subset of the inputs. The weight constraint scheme directly controls the Lipschitz constant of the neural network and thus provides the additional benefit of robustness. Compared to currently existing techniques used for monotonicity, our method is simpler in implementation and in theory foundations, has negligible computational overhead, is guaranteed to produce monotonic dependence, and is highly expressive. We show how the algorithm is used to train powerful, robust, and interpretable discriminators that achieve competitive performance compared to current state-of-the-art methods across various benchmarks, from social applications to the classification of the decays of subatomic particles produced at the CERN Large Hadron Collider.

## 1 INTRODUCTION

The need to model functions that are monotonic in a subset of their inputs is prevalent in many ML applications. Enforcing monotonic behaviour can help improve generalization capabilities (Milani Fard et al., 2016; You et al., 2017) and assist with interpretation of the decision-making process of the neural network (Nguyen & Martínez, 2019). Real world scenarios include various applications with fairness, interpretability, and security aspects. Examples can be found in the natural sciences and in many social applications. Monotonic dependence of a model output on a certain feature in the input can be informative of how an algorithm works—and in some cases is essential for real-word usage. For instance, a good recommender engine will favor the product with a high number of reviews over another with fewer but otherwise identical reviews (*ceteris paribus*). The same applies for systems that assess health risk, evaluate the likelihood of recidivism, rank applicants, filter inappropriate content, *etc*.

In addition, robustness to small perturbations in the input is a desirable property for models deployed in real world applications. In particular, when they are used to inform decisions that directly affect human actors—or where the consequences of making an unexpected and unwanted decision could be extremely costly. The continued existence of adversarial methods is a good example for the possibility of malicious attacks on current algorithms (Akhtar et al., 2021). A natural way of ensuring the robustness of a model is to constrain its Lipschitz constant. To this end, we recently developed an architecture whose Lipschitz constant is constrained by design using layer-wise normalization which allows the architecture to be more expressive than the current state-of-the-art with stable and fast training (Kitouni et al., 2021). Our algorithm has been adopted to classify the decays of subatomic particles produced at the CERN Large Hadron Collider in the real-time data-processing system of the LHCb experiment, which was our original motivation for developing this novel architecture.

---

[*]Equal contribution

In this paper, we present expressive monotonic Lipschitz networks. This new class of architectures employs the Lipschitz bounded networks from Kitouni et al. (2021) along with residual connections to implement monotonic dependence in any subset of the inputs by construction. It also provides exact robustness guarantees while keeping the constraints minimal such that it remains a universal approximator of Lipschitz continuous monotonic functions. We show how the algorithm is used to train powerful, robust, and interpretable discriminators that achieve competitive performance compared to current state-of-the-art methods across various benchmarks, from social applications to its original target application: the classification of the decays of subatomic particles produced at the CERN Large Hadron Collider.

## 2 RELATED WORK

Prior work in the field of monotonic models can be split into two major categories.

- **Built-in and constrained monotonic architectures**: Examples of this category include Deep Lattice Networks (You et al., 2017) and networks in which all weights are constrained to have the same sign (Sill, 1998). The major drawbacks of most implementations of constrained architectures are a lack of expressiveness or poor performance due to superfluous complexity.

- **Heuristic and regularized architectures (with or without certification)**: Examples of such methods include Sill & Abu-Mostafa (1996) and Gupta et al., which penalizes point-wise negative gradients on the training sample. This method works on arbitrary architectures and retains much expressive power but offers no guarantees as to the monotonicity of the trained model. Another similar method is Liu et al. (2020), which relies on Mixed Integer Linear Programming to certify the monotonicity of piece-wise linear architectures. The method uses a heuristic regularization to penalize the non-monotonicty of the model on points sampled uniformly in the domain during training. The procedure is repeated with increasing regularization strength until the model passes the certification. This iteration can be expensive and while this method is more flexible than the constrained architectures (valid for MLPs with piece-wise linear activations), the computational overhead of the certification process can be prohibitively expensive. Similarly, Sivaraman et al. (2020) propose guaranteed monotonicity for standard ReLU networks by letting a Satisfiability Modulo Theories (SMT) solver find counterexamples to the monotonicity definition and adjust the prediction in the inference process such that monotonicity is guaranteed. However, this approach requires queries to the SMT solver during inference time for each monotonic feature, and the computation time scales harshly with the number of monotonic features and the model size (see Figure 3 and 4 in Sivaraman et al. (2020)).

Our architecture falls into the first category. However, we overcome both main drawbacks: lack of expressiveness and impractical complexity. Other related works appear in the context of monotonic functions for normalizing flows, where monotonicity is a key ingredient to enforce invertibility (De Cao et al., 2020; Huang et al., 2018; Behrmann et al., 2019; Wehenkel & Louppe, 2019).

## 3 METHODS

The goal is to develop a neural network architecture representing a vector-valued function

$$f : \mathbb{R}^d \to \mathbb{R}^n, \quad d, n \in \mathbb{N}, \tag{1}$$

that is provably monotonic in any subset of its inputs. We first define a few ingredients.

**Definition 3.1** (Monotonicity). Let $\boldsymbol{x} \in \mathbb{R}^d$, $\boldsymbol{x}_{\mathbb{S}} \equiv \mathbf{1}_{\mathbb{S}} \odot \boldsymbol{x}$, and the Hadamard product of $\boldsymbol{x}$ with the indicator vector $\mathbf{1}_{\mathbb{S}}(i) = 1$ if $i \in \mathbb{S}$ and 0 otherwise for a subset $\mathbb{S} \subseteq \{1, \cdots, d\}$.

We say that outputs $\mathbb{Q} \subseteq \{1, \cdots, n\}$ of $f$ are monotonically increasing in features $\mathbb{S}$ if

$$f(\boldsymbol{x}'_{\mathbb{S}} + \boldsymbol{x}_{\bar{\mathbb{S}}})_i \leq f(\boldsymbol{x}_{\mathbb{S}} + \boldsymbol{x}_{\bar{\mathbb{S}}})_i \quad \forall i \in \mathbb{Q} \text{ and } \forall \boldsymbol{x}'_{\mathbb{S}} \leq \boldsymbol{x}_{\mathbb{S}}, \tag{2}$$

where $\bar{\mathbb{S}}$ denotes the complement of $\mathbb{S}$ and the inequality on the right uses the product (or component-wise) order.

**Definition 3.2** (Lip$^p$ function). $g : \mathbb{R}^d \to \mathbb{R}^n$ is Lip$^p$ if it is Lipschitz continuous with respect to the $L^p$ norm in every output dimension, *i.e.*,

$$||g(\boldsymbol{x}) - g(\boldsymbol{y})||_\infty \leq \lambda ||\boldsymbol{x} - \boldsymbol{y}||_p \quad \forall \boldsymbol{x}, \boldsymbol{y} \in \mathbb{R}^n. \tag{3}$$

### 3.1 LIPSCHITZ MONOTONIC NETWORKS (LMN)

We will henceforth and without loss of generality only consider scalar-valued functions ($n = 1$). We start with a model $g(\boldsymbol{x})$ that is Lip[1] with Lipschitz constant $\lambda$. Note that the choice of $p = 1$ is crucial for decoupling the magnitudes of the directional derivatives in the monotonic features. More details on this can be found below and in Figure 1. The 1-norm has the convenient side effect that we can tune the robustness requirement for each input individually.

With a model $g(\boldsymbol{x})$ we can define an architecture with built-in monotonicity by adding a term that has directional derivative $\lambda$ for each coordinate in $\mathbb{S}$:

$$f(\boldsymbol{x}) = g(\boldsymbol{x}) + \lambda(\mathbf{1}_{\mathbb{S}} \cdot \boldsymbol{x}) = g(\boldsymbol{x}) + \lambda \sum_{i \in \mathbb{S}} x_i. \tag{4}$$

This residual connection $\lambda(\mathbf{1}_{\mathbb{S}} \cdot \boldsymbol{x})$ enforces monotonicity in the input subset $\boldsymbol{x}_{\mathbb{S}}$:

$$\frac{\partial g}{\partial x_i} \in [-\lambda, \lambda], \ \ \forall\, i \in \mathbb{N}_{1:n} \tag{5}$$

$$\Rightarrow \frac{\partial f}{\partial x_i} = \frac{\partial g}{\partial x_i} + \lambda \geq 0 \ \forall \boldsymbol{x} \in \mathbb{R}^n, i \in \mathbb{S}. \tag{6}$$

**The importance of the norm choice** The construction presented here does not work with $p \neq 1$ constraints because dependencies between the partial derivatives may be introduced, see Figure 1. The $p = 1$-norm is the only norm that bounds the gradient within the green square and, crucially, allows the directional derivatives to be as large as $2\lambda$ independently. When shifting the constraints by introducing the linear term, the green square allows for all possible gradient configurations, given that we can choose $\lambda$ freely. As a counter example, the red circle, corresponding to $p = 2$ constraints, prohibits important areas in the configuration space.

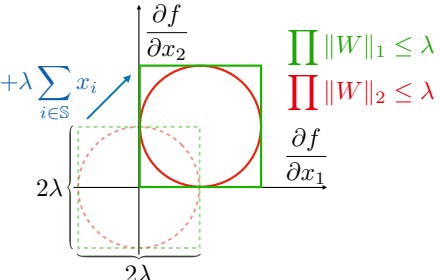

Figure 1: $p$-norm constrained gradients showing (red) $p = 2$ and (green) $p = 1$. The gradient of a function $g(\boldsymbol{x})$ that is Lip[p=2] resides within the dashed red line. For a Lip[p=1] function, the boundary is the green dashed line. Note that $\boldsymbol{x}$ is taken to be a row vector. The residual connection (in blue) effectively shifts the possible gradients to strictly positive values and thus enforces monotonicity. Note how the red solid circle does not include all possible gradient configurations. For instance, it does not allow for very small gradients in both inputs, whereas the green square includes all configurations, up to an element-wise maximum of $2\lambda$.

To be able to represent all monotonic Lip[1] functions with $2\lambda$ Lipschitz constant, the construction of $g(\boldsymbol{x})$ needs to be a universal approximator of Lip[1] functions. In the next section, we will discuss possible architectures for this task.

### 3.2 LIP[p=1] APPROXIMATORS

Our goal is to construct a universal approximator of Lip[1] functions, *i.e.*, we would like the hypothesis class to have two properties:

1. It always satisfies Eq. 3, *i.e.*, be Lip[1].
2. It is able to fit all possible Lip[1] functions. In particular, the bound in Eq. 3 needs to be attainable $\forall\, \boldsymbol{x}, \boldsymbol{y}$.

**Lip[1] constrained models** To satisfy the first requirement, fully connected networks can be Lipschitz bounded by constraining the matrix norm of all weight matrices (Kitouni et al., 2021; Gouk et al., 2020; Miyato et al., 2018). We recursively define the layer $l$ of the fully connected network of depth $D$ with activation $\sigma$ as

$$z^l = \sigma(z^{l-1})W^l + b^l, \tag{7}$$

where $z^0 = x$ is the input and $f(x) = z^D$ is the output of the neural network. It follows that $g(x)$ satisfies Eq. 3 if

$$\prod_{i=1}^{D} \|W^i\|_1 \leq \lambda, \tag{8}$$

and $\sigma$ has a Lipschitz constant less than or equal to 1. There are multiple ways to enforce Eq. 8. Two existing possibilities that involve scaling by the operator norm of the weight matrix (Gouk et al., 2020) are:

$$W^i \to W'^i = \lambda^{1/D} \frac{W^i}{\max(1, \|W^i\|_1)} \qquad \text{or} \qquad W^i \to W'^i = \frac{W^i}{\max(1, \lambda^{-1/D} \cdot \|W^i\|_1)}. \tag{9}$$

In our studies, the latter variant seems to train slightly better. However, in some cases it might be useful to use the former to avoid the scale imbalance between the neural network's output and the residual connection used to induce monotonicity. We note that in order to satisfy Eq. 8, it is not necessary to divide the entire matrix by its 1-norm. It is sufficient to ensure that the absolute sum over each column is constrained:

$$W^i \to W'^i = W^i \text{diag} \left( \frac{1}{\max \left( 1, \lambda^{-1/D} \cdot \sum_j |W_{jk}^i| \right)} \right). \tag{10}$$

This novel normalization scheme tends to give even better training results in practice, because the constraint is applied in each column individually. This reduces correlations of constraints, in particular, if a column saturates the bound on the norm, the other columns are not impacted. While Eq. 10 may not be suitable as a general-purpose scheme, *e.g.* it would not work in convolutional networks, its performance in training in our analysis motivates its use in fully connected architectures and further study of this approach in future work.

In addition, the constraints in Eq. 9 and Eq. 10 can be applied in different ways. For example, one could normalize the weights directly before each call such that the induced gradients are propagated through the network like in Miyato et al. (2018). While one could come up with toy examples for which propagating the gradients in this way hurts training, it appears that this approach is what usually is implemented for spectral norm in PyTorch and TensorFlow (Miyato et al., 2018) . Alternatively, the constraint could be applied by projecting any infeasible parameter values back into the set of feasible matrices after each gradient update as in Algorithm 2 of Gouk et al. (2020).

Constraining according to Eq. 8 is not the only way to enforce Lip[1]. Anil et al. (2019) provide an alternative normalization scheme:

$$\|W^1\|_{1,\infty} \cdot \prod_{i=2}^{m} \|W^i\|_\infty \leq \lambda \tag{11}$$

Similarly to how the 1-norm of a matrix is a column-wise maximum, the $\infty$-norm of a matrix is determined by the maximum 1-norm of all rows and $\|W\|_{1,\infty}$ simply equals the maximum absolute value of an element in the matrix. Therefore, normalization schemes similar to Eq. 10, can be employed to enforce the constraints in Eq. 11 by replacing the column-wise normalization with a row- or element-wise normalization where appropriate.

**Preserving expressive power** Guaranteeing that the model is Lipschitz bounded is not sufficient, it must also able to saturate the bound to be able to model all possible Lip[1] functions. Some Lipschitz network architectures, *e.g.* Miyato et al. (2018), tend to over constrain the model such that it cannot fit all Lip[1] functions due to *gradient attenuation*. For many problems this is a rather theoretical issue. However, it becomes a practical problem for the monotonic architecture since it often works

on the edges of its constraints, for instance when partial derivatives close to zero are required, see Figure 1. As a simple example, the authors of Huster et al. (2018) showed that ReLU networks are unable to fit the function $f(x) = |x|$ if the layers are norm-constrained with $\lambda = 1$. The reason lies in fact that ReLU, and most other commonly used activations, do not have unit gradient with respect to the inputs over their entire domain. While monotonic element-wise activations like ReLU cannot have unit gradient almost everywhere without being exactly linear, the authors of Anil et al. (2019) explore activations that introduce non-linearities by reordering elements of the input vector. They propose **GroupSort** as an alternative to point-wise activations, and it is defined as follows:

$$\sigma_G(\boldsymbol{x}) = \text{sort}_{1:G}(\boldsymbol{x}_{1:G}) + \text{sort}_{G+1:2G}(\boldsymbol{x}_{G+1:2G}) + \dots$$
$$= \sum_{i=0}^{n/G-1} \text{sort}_{iG+1:(i+1)G}(\boldsymbol{x}_{iG+1:(i+1)G}), \tag{12}$$

where $\boldsymbol{x} \in \mathbb{R}^n$, $\boldsymbol{x}_{i:j} = \mathbf{1}_{i:j} \odot \boldsymbol{x}$, and $\text{sort}_{i:j}$ orders the elements of a vector from indices $i$ to $j$ and leaves the other elements in place. This activation sorts an input vector in chunks (groups) of a fixed size $G$. The GroupSort operation has a gradient of unity with respect to every input, giving architectures constrained with Eq. 8 greatly increased expressive power. In fact, Anil et al. (2019) prove that GroupSort networks with the normalization scheme in Eq. 11 are universal approximators of Lip[1] functions. Therefore, these networks fulfill the two requirements outlined in the beginning of this section.

For universal approximation to be possible, the activation function used needs to be gradient norm preserving (GNP), *i.e.,* have gradient 1 almost everywhere. Householder activations are another instance of GNP activations of which GroupSort-2 is a special case (Singla et al., 2021). The Householder activation is defined as follows:

$$\sigma(\boldsymbol{z}) = \begin{cases} \boldsymbol{z} & \boldsymbol{z}\boldsymbol{v} > 0 \\ \boldsymbol{z}(\mathbf{I} - 2\boldsymbol{v}\boldsymbol{v}^T) & \boldsymbol{z}\boldsymbol{v} \leq 0 \end{cases} \tag{13}$$

Here, $\boldsymbol{z}$ is the preactivation row vector and $\boldsymbol{v}$ is any column unit vector. Householder Lipschitz Networks naturally inherit the universal approximation property.

In summary, we have constructed a neural network architecture $f(\boldsymbol{x})$ via Eq. 4 that can provably approximate all monotonic Lipschitz bounded functions. The Lipschitz constant of the model can be increased arbitrarily by controlling the parameter $\lambda$ in our construction.

## 4 EXPERIMENTS

"Beware of bugs in the above code, I have only proved it correct, not tried it" (Knuth). In the spirit of Donald Knuth, in this section we test our algorithm on many different domains to show that it works well in practice and gives competitive results, as should be expected from a universal approximator.

### 4.1 TOY EXAMPLE

Figure 2 shows a toy example where both a monotonic and an unconstrained network are trained to regress on a noisy one-dimensional dataset. The true underlying model used here is monotonic, though an added heteroskedastic Gaussian noise term can obscure this in any realization. As can be seen in Figure 2, no matter how the data are distributed at the edge of the support, the monotonic Lipschitz network is always non-decreasing outside of the support as guaranteed by our architecture. Such out-of-distribution guarantees can be extremely valuable in cases where domain knowledge dictates monotonic behavior is either required or desirable.

### 4.2 REAL-TIME DECISION-MAKING AT 40 MHz AT THE LHC

Because many physical systems are modeled with well-known theoretical frameworks that dictate the properties of the system, monotonicity can be a crucial inductive bias in the physical sciences. For instance, modeling enthalpy, a thermodynamic quantity measuring the total heat content of a system, in a simulator requires a monotonic function of temperature for fixed pressure (as is known from basic physical principles).

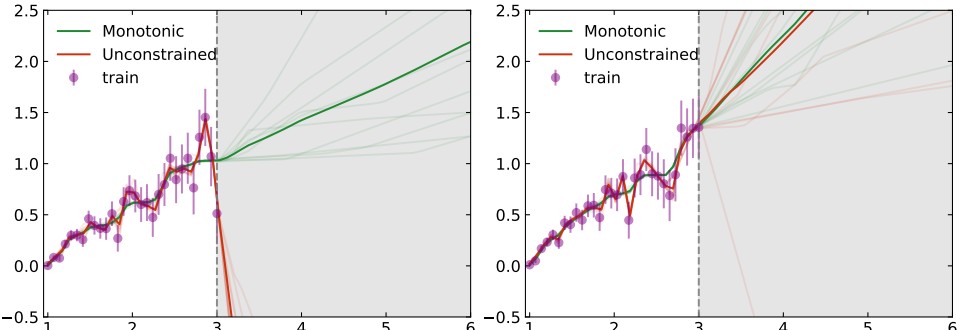

Figure 2: Our monotonic architecture (green) and an unconstrained network (red) trained on two realizations (purple data points) of a one dimensional dataset. The shaded regions are where training data were absent. Each model is trained using 10 random initialization seeds. The dark lines are averages over the seeds, which are each shown as light lines. The unconstrained models exhibit overfitting of the noise, non-monotonic behavior, and highly undesirable and unpredictable results when extrapolating beyond the region occupied by the training data. Conversely, the monotonic Lipschitz models are always monotonic, even in scenarios where the noise is strongly suggestive of non-monotonic behavior. In addition, the Lipschitz constraint produces much smoother models.

In this section, we describe a real-world physics application which requires monotonicity in certain features—and robustness in all of them. The algorithm described here has, in fact, been implemented by a high-energy particle physics experiment at the European Center for Nuclear Research (CERN), and is actively being used to collect data at the Large Hadron Collider (LHC) in 2022, where high-energy proton-proton collisions occur at 40 MHz.

The sensor arrays of the LHC experiments produce data at a rate of over 100 TB/s. Drastic data-reduction is performed by custom-built read-out electronics; however, the annual data volumes are still $O(100)$ exabytes, which cannot be put into permanent storage. Therefore, each LHC experiment processes its data in real time, deciding which proton-proton collision events should be kept and which should be discarded permanently; this is referred to as *triggering* in particle physics. To be suitable for use in trigger systems, classification algorithms must be robust against the impact of experimental instabilities that occur during data taking—and deficiencies in simulated training samples. Our training samples cannot possibly account for the unknown new physics that we hope to learn by performing the experiments!

A ubiquitous inductive bias at the LHC is that outlier collision events are more interesting, since we are looking for physics that has never been observed before. However, uninteresting outliers are frequently caused by experimental imperfections, many of which are included and labeled as background in training. Conversely, it is not possible to include the set of all possible interesting outliers *a priori* in the training. A solution to this problem is to implement *outliers are better* directly using our expressive monotonic Lipschitz architecture from Section 3.

Our architecture was originally developed for the task of classifying the decays of heavy-flavor particles produced at the LHC. These are bound states containing a beauty or charm quark that travel an observable distance $\mathcal{O}(1\,\mathrm{cm})$ before decaying due to their (relatively) long lifetimes. This example uses a dataset of simulated proton-proton ($pp$) collisions in the LHCb detector. Charged particles recorded by LHCb are combined pairwise into decay-vertex (DV) candidates. The task concerns discriminating DV candidates corresponding to heavy-flavor decays from all other sources. Heavy-flavor DVs typically have substantial separation from the $pp$ collision point, due to the relatively long heavy-flavor particle lifetimes, and large transverse momenta, $p_\mathrm{T}$, of the component particles, due to the large heavy-flavor particle masses. The main sources of background DVs, described in Kitouni et al. (2021), mostly have small displacement and small $p_\mathrm{T}$, though unfortunately they can also have extremely large values of both displacement and momentum.

Figure 3 shows a simplified version of this problem using only the two most-powerful inputs. Our inductive bias requires a monotonic increasing response in both features (detailed discussion motivating this bias can be found in Kitouni et al. (2021)). We see that an unconstrained neural network

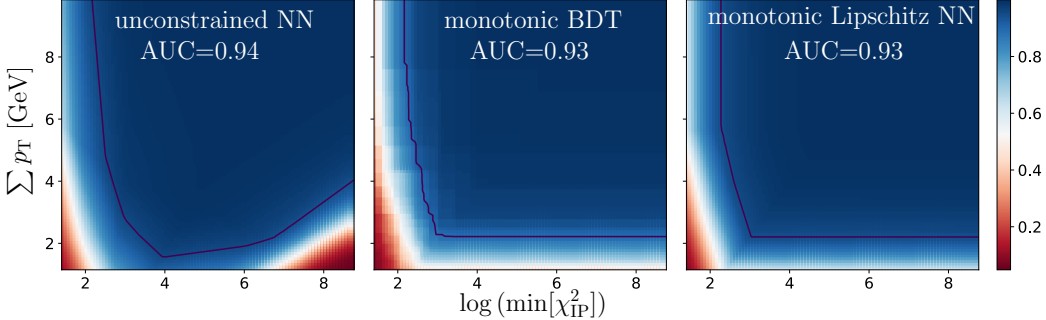

Figure 3: From Kitouni et al. (2021): Simplified version of the heavy-quark selection problem using only two inputs, which permits displaying the response everywhere in the feature space; shown here as a heat map with more signal-like (background-like) regions colored blue (red). The dark solid line shows the decision boundary (upper right regions are selected). Shown are (left) a standard fully connected neural network, (middle) a monotonic BDT, and (right) our architecture. The quantities shown on the horiztonal and vertical axes are related to how long the particle lived before decaying and how massive the particle was, respectively.

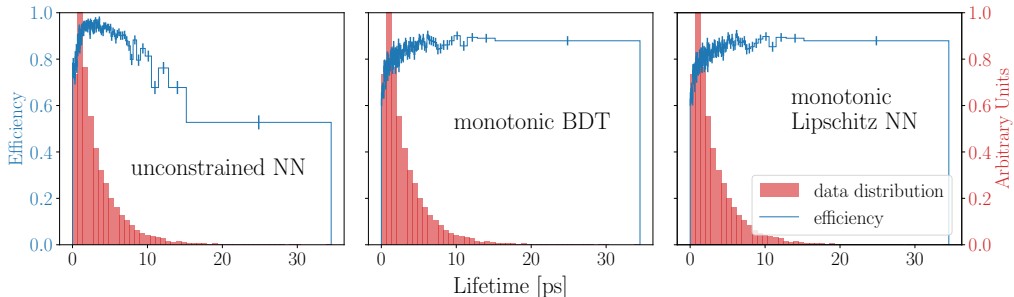

Figure 4: From Kitouni et al. (2021): True positive rate (efficiency) of each model shown in Figure 3 versus the proper lifetime of the decaying heavy-quark particle selected. The monotonic models produce a nearly uniform efficiency above a few picoseconds at the expense of a few percent lifetime-integrated efficiency. Such a trade off is desirable as explained in the text.

rejects DVs with increasing larger displacements (lower right corner), and that this leads to a decrease of the signal efficiency (true positive rate) for large lifetimes. The unconstrained model violates our inductive bias. Figures 3 and 4 show that a monotonic BDT (Auguste et al., 2020) approach works here. However, the jagged decision boundary can cause problems in subsequent analysis of the data. Figure 3 also shows that our novel approach from Section 3 successfully produces a smooth and monotonic response, and Figure 4 shows that this provides the monotonic lifetime dependence we desire in the efficiency.

In addition, we note that the added benefit of guaranteed Lipschitz robustness is a major advantage for many real world applications. Specifically for particle physicists, this kind of robustness directly translates to important guarantees when considering experimental instabilities.

Due to the simplicity and practicality of our method, the LHCb experiment is now using the proposed architecture for real-time data selection at a data rate of about 40 Tbit/s.

### 4.3    PUBLIC DATASETS WITH MONOTONIC DEPENDENCE

In this section, we follow as closely as possible the experiments done in Liu et al. (2020), and some experiments done in Sivaraman et al. (2020) to be able to directly compare to state-of-the-art monotonic architectures. Liu et al. (2020) studied monotonic architectures on four different datasets: COMPAS (Larson & Kirchner, 2016), BlogFeedback (Buza, 2014), LoanDefaulter (Kaggle, 2015),

and ChestXRay (Wang et al., 2017). From Sivaraman et al. (2020) we compare against one regression and one classification task: AutoMPG (Dua & Graff, 2017) and HeartDisease (Gennari et al., 1989). Results are shown in Table 1.

**COMPAS** (Correctional Offender Management Profiling for Alternative Sanctions) refers to a commercial algorithm used by judges and police officers to determine the likelihood of reoffense. Larson & Kirchner (2016) discusses that the algorithm is racially biased and provides a dataset from a two-year study of the real-world performance of COMPAS. The task here is to predict the reoffense probability within the next two years. The dataset has 13 features, 4 of which have a monotonic inductive bias, and contains a total of 6172 data points.

**BlogFeedBack** This dataset contains 54270 data points with 276 dimensions describing blog posts. The task is to predict the number of comments following the post publication within 24 hours. 8 of the features have a monotonic inductive bias. Just like Liu et al. (2020), we also only consider the 90% of the data points with the smallest targets so as to not let the RMSE be dominated by outliers.

**LoanDefaulter** The version of this dataset available on Kaggle was updated on a yearly basis up to 2015. Kaggle (2015) contains a link that is, we believe, a superset of the data used in Liu et al. (2020). Luckily, the authors have shared with us the exact version of the dataset they used in their studies for an appropriate comparison. The data is organized in 28 features and the task is to determine loan defaulters. The classification score should be monotonic in 5 features: non-decreasing in number of public record bankruptcies and Debt-to-Income ratio, non-increasing in credit score, length of employment and annual income.

**ChestXRay** This dataset contains tabular data and images of patients with diseases that are visible in a chest x-ray. The task is to predict whether or not the patient has such a disease. Just like Liu et al. (2020), we send the image through an ImageNet-pretrained ResNet18 (He et al., 2016). The penultimate layer output concatenated with tabular data acts as input to the monotonic architecture. Two of the four tabular features are monotonic. In the bottom right table in 1, there are two entries for our architecture. The *E-E* entry refers to end-to-end training with ResNet18, whereas the other experiment fixes the ResNet weights.

**AutoMPG** (Dua & Graff, 2017) This is a dataset containing 398 examples of cars, described by 7 numerical features and the model name. The target, MPG, is monotonically decreasing with 3 of the features. The name is not used as a feature.

**HeartDisease** (Gennari et al., 1989) is a dataset of patients, described by 13 features. The task is to determine whether or not the patient has heart disease.

As can be seen in Table 1, our Lipschitz monotonic networks perform competitively or better than the state-of-the-art on all benchmarks we tried.

It is also immediately apparent that our architecture is highly expressive. We manage to train tiny networks with few parameters while still achieving competitive performance. Given that some of these datasets have a significant number of features compared to our chosen network width, most parameters are in the weights of the first layer. We manage to build and train even smaller networks with better generalization performance when taking only a few important features. These networks are denoted with **mini** in Table 1. Because all of the presented architectures are small in size, we show practical finite sample expressiveness for harder tasks and larger networks by achieving 100% training accuracy on MINST, CIFAR-10, and CIFAR-100 with real and random labels as well as an augmented version (i.e. with an additional monotonic feature added artificially) of CIFAR100 in Appendix A.

## 5 LIMITATIONS

We are working on improving the architecture as follows: First, common initialization techniques are not optimal for weight-normed networks (Arpit et al., 2019). Simple modifications to the weight

**COMPAS**

| Method | Parameters | ⇈ Test Acc |
|---|---|---|
| Certified | 23112 | $(68.8 \pm 0.2)\%$ |
| **LMN** | **37** | $(\mathbf{69.3 \pm 0.1})\%$ |

**BlogFeedback**

| Method | Parameters | ⇊ RMSE |
|---|---|---|
| Certified | 8492 | $.158 \pm .001$ |
| **LMN** | **2225** | $\mathbf{.160 \pm .001}$ |
| **LMN mini** | **177** | $\mathbf{.155 \pm .001}$ |

**LoanDefaulter**

| Method | Parameters | ⇈ Test Acc |
|---|---|---|
| Certified | 8502 | $(65.2 \pm 0.1)\%$ |
| **LMN** | **753** | $(\mathbf{65.44 \pm 0.03})\%$ |
| **LMN mini** | **69** | $(\mathbf{65.28 \pm 0.01})\%$ |

**ChestXRay**

| Method | Parameters | ⇈ Test Acc |
|---|---|---|
| Certified | 12792 | $(62.3 \pm 0.2)\%$ |
| Certified E-E | 12792 | $(66.3 \pm 1.0)\%$ |
| **LMN** | **1043** | $(\mathbf{67.6 \pm 0.6})\%$ |
| **LMN E-E** | **1043** | $(\mathbf{70.0 \pm 1.4})\%$ |

**Heart Disease**

| Method | ⇈ Test Acc |
|---|---|
| COMET | $(86 \pm 3)\%$ |
| **LMN** | $(\mathbf{89.6 \pm 1.9})\%$ |

**Auto MPG**

| Method | ⇊ MSE |
|---|---|
| COMET | $(8.81 \pm 1.81)\%$ |
| **LMN** | $(\mathbf{7.58 \pm 1.2})\%$ |

Table 1: We compare our method (in bold) against state-of-the-art monotonic models (we only show the best) on a variety of benchmarks. The performance numbers for other techniques were taken from Liu et al. (2020) and Sivaraman et al. (2020). In the ChestXRay experiment, we train one model with frozen ResNet18 weights (second to last) and another with end-to-end training (last). While our models can generally get quite small, we can achieve even smaller models when only taking a subset of all the features. These models are denoted with "mini".

initialization might aid convergence, especially for large Lipschitz parameters. Secondly, we are currently constrained to activation functions that have a gradient norm of 1 over their entire domain, such as **GroupSort**, to ensure universal approximation, see Anil et al. (2019). We will explore other options in the future. Lastly, there is not yet a proof for universal approximation for the architecture described in Eq. 8. However, it appears from empirical investigation that the networks do approximate universally, as we have yet to find a function that could not be approximated well enough with a deep enough network. We do not consider this a major drawback, as the construction in Eq. 11 does approximate universally, see Anil et al. (2019). Note that none of these limitations have any visible impact on the performance of the experiments in Section 4.

## 6 CONCLUSION AND FUTURE WORK

We presented an architecture that provably approximates Lipschitz continuous and partially monotonic functions. Monotonic dependence is enforced via an end-to-end residual connection to a minimally Lip[1] constrained fully connected neural network. This method is simple to implement, has negligible computational overhead, and gives stronger guarantees than regularized models. Our architecture achieves competitive results with respect to current state-of-the-art monotonic architectures, even when using a tiny number of parameters, and has the additional benefit of guaranteed robustness due to its known Lipschitz constant. For future directions of this line of research, we plan to tackle the problems outlined in the limitation section, especially improving initialization of weight-normed networks.

## 7 REPRODUCIBILITY STATEMENT

All experiments with public datasets are reproducible with the code provided at `https://github.com/niklasnolte/monotonic_tests`. The experiments in Section 4.2 were made with data that is not publicly available. The code to reproduce those experiments can be found under `https://github.com/niklasnolte/HLT_2Track` and the data will be made available in later years at the discretion of the LHCb collaboration.

ACKNOWLEDGMENTS

This work was supported by NSF grant PHY-2019786 (The NSF AI Institute for Artificial Intelligence and Fundamental Interactions, http://iaifi.org/).

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

## A  EXPRESSIVE POWER OF THE ARCHITECTURE

Robust architectures like Lipschitz constrained networks are often believed to be much less expressive than their unconstrained counterparts Huster et al. (2019). Here we show that our architecture is capable of (over)fitting complex decision boundaries even on random labels in a setup simular to Zhang et al. (2021).

We show the finite sample expressiveness of the architecture in `https://github.com/okitouni/Lipschitz-network-bench` by fitting MNIST, CIFAR10, CIFAR100 with normal and random labels to 100% training accuracy. We also train on CIFAR100 with an additional "goodness" feature $x \in [0, 1]$ to showcase the monotonicity aspect of the architecture. This dataset is referred to as CIFAR101 below. The synthetic monotonicity problem is currently implemented such that samples with values above a critical threshold in the goodness feature $x > x_{\text{crit}}$ are labeled 0. An alternative implementation is to take label 0 with probability $x$ and keep the original label (or assign a random one) with probability $1 - x$. Table 2 summarizes the setup used for training. We use Adam with default hyper-parameters im all experiments.

| Task | Width | Depth | LR | EPOCHS | Batchsize | Loss |
|---|---|---|---|---|---|---|
| MNIST | 1024 | 3 | $10^{-5}$ | $10^5$ | ALL | $CE(\tau = 256)$ |
| CIFAR10 | 1024 | 3 | $10^{-5}$ | $10^5$ | ALL | $CE(\tau = 256)$ |
| CIFAR100/101 | 1024 | 3 | $10^{-5}$ | $10^5$ | ALL | $CE(\tau = 256)$ |

Table 2: Training MNIST and CIFAR10/100 to 100% training accuracy with Lipschitz networks.

