# OpenReview forum: "Expressive Monotonic Neural Networks"
_ICLR.cc/2023/Conference — ICLR 2023 poster_

### Official Review · Reviewer_8Geq · 2022-10-21

**Confidence:** 4
**Clarity, Quality, Novelty And Reproducibility:** (see Strengths and Weaknesses)
**Correctness:** 4
**Technical Novelty And Significance:** 3
**Empirical Novelty And Significance:** 3
**Recommendation:** 8

**Strength And Weaknesses:**

### Strengths

- The paper relies on a simple (but seemingly novel) idea: by adding a residual connection to a weight-constrained Lipschitz network, one can obtain a network that inherits the expressiveness of the Lipschitz network, while the bound of the Lipschitz norm can be used to specify the strength of the residual connection to ensure monotonicity. This is simpler and is certainly more powerful than two existing ideas in the field (ensure positive weight across the nodes of a layer, or penalize the negative gradient in the samples).
- The main ideas of the work were developed based on well-founded rationales; the technical details (such as choice of the norm, or how datasets are obtained and analyzed) are carefully designed. The paper is well-written with a sufficient literature review. Visual information is intuitive.
- The problem studied in this paper is an important and of great interest to a broad audience in the field. Its impact on applied science is highlighted by the motivating example (real-time decision-making at high frequency at the Large Hadron Collider). As stated in the paper, the LHCb experiment is now using the proposed architecture for real-time data selection.
- The design of the experiments is very careful and rigorous. The results showcase that the new approach could outperform state-of-the-art methods across various benchmarks previously studied in the field.

### Weaknesses

- None noted. However, it could be argued that the main technical part of the work (technical structure, expressiveness, and universal approximation of Lipschitz networks) are all pre-existing results in the field and that the technical and theoretical contributions of the work are minimal.

**Summary Of The Paper:**

The paper constructs a novel type of architecture for monotonic neural networks by adding a single residual connection to an expressive (weight-constrained) Lipschitz network. The resulting network architecture is simple in implementation and theory foundations, robust, and highly expressive. The algorithm is compared to state-of-the-art methods across various benchmarks.

**Summary Of The Review:**

The paper addresses an important question. The approach is novel and of broad interest. The proposed structure is better designed and outperforms pre-existing approaches in constructing monotonic networks.

---

> ### Author Response · Authors · 2022-11-11
> **Response to 8Geq**
>
> Dear Reviewer 8Geq,
>
> Thank your favorable and thorough review!

---

### Official Review · Reviewer_rnj5 · 2022-10-21

**Confidence:** 4
**Correctness:** 4
**Technical Novelty And Significance:** 3
**Empirical Novelty And Significance:** 3
**Recommendation:** 8

**Clarity, Quality, Novelty And Reproducibility:**

Very clear overall. I could understand it with minimal difficulty. Seems novel/original, as far as I know.

Nits:
* Many of the citations are incorrectly formatted (need parentheses around them).
* "In this article" should be "In this paper"
* "Fig 3.1" should be "Fig 1"

**Strength And Weaknesses:**

Strengths:
* Achieves guaranteed monotonicity with a relatively simple function class that is easy and efficient to implement and train.
* Side benefit of a regularized function class, due to the Lip_1 constraint. But this may be a weakness when highly-flexible function classes are desirable without needing to resort to very deep models.

Weaknesses:
* Uses a somewhat unusual activation function, "GroupSort". I wonder if there are alternatives that can satisfy the conditions for monotonicity and are more similar to more commonly-used activation functions.

**Summary Of The Paper:**

The paper introduces a novel method for learning monotonic neural networks, meaning that its output can be constrained and guaranteed to always increase (or decrease) with respect to any subset of its inputs. It accomplishes this by starting with a standard feedforward architecture whose weights and activation function are bounded so that the function has a Lipschitz constant of 1, and then they simply add (or subtract) a residual connection from the monotonic input directly to the network's output, guaranteeing that the partial derivative w.r.t. the monotonic input is always positive (or negative).

The authors show empirically that this architecture performs favorably compared to an unconstrained network, as well as compared to a variety of prior work on monotonic machine-learned function classes.

**Summary Of The Review:**

A very nice, clear, idea for training monotonic functions that seems relatively easy to implement and very effective.

---

> ### Author Response · Authors · 2022-11-11
> **Response to rnj5**
>
> Dear reviewer rnj5,
>
> Thank you for your constructive review, in particular, for pointing out the inconsistencies in the citations. We have now edited the submission to have consistent citations.
>
> We have also implemented your other suggestions and added a paragraph (in blue) on alternative activation functions that can be used in this context. For a more detailed explanation as to why standard activation functions cannot be used, please see bullet point 2 in the comment to reviewer tc6i (https://openreview.net/forum?id=w2P7fMy_RH&noteId=WQPnUw9EnE1) as well as the revised submission (in particular Appendix A).

---

### Official Review · Reviewer_tc6i · 2022-10-24

**Confidence:** 4
**Correctness:** 2
**Technical Novelty And Significance:** 2
**Empirical Novelty And Significance:** 3
**Recommendation:** 3

**Clarity, Quality, Novelty And Reproducibility:**

Clarity is good.

Quality is fair.

Novelty is poor. Please see the comments above.

Reproducibility is good.


**Details Of Ethics Concerns:**

It breaks double blind review policy.

**Strength And Weaknesses:**

This paper contains the following statements that breaks the double blind review policy. This may warrant rejection without review and I'll leave the decision to the Area Chair.

-- "Our algorithm has been adopted to classify the decays of subatomic particles produced at the CERN Large Hadron Collider in the real-time data-processing system of the LHCb experiment, which was our original motivation for developing this novel architecture."

-- "The algorithm described here has, in fact, been implemented by a high-energy particle physics experiment at the European Center for Nuclear Research (CERN), and is actively being used to collect data at the Large Hadron Collider (LHC) in 2022, where highenergy proton-proton collisions occur at 40 MHz."

-- "Due to the simplicity and practicality of our method, the LHCb experiment is now using the proposed architecture for real-time data selection at a data rate of about 40 Tbit/s."

-- "Those data will be made available in later years at the discretion of the LHCb collaboration."


Having said the above, I'll provide a brief review of the proposed technique anyway.

The motivation of this work is meaningful. The monotonicity guarantee is correct.

However, whether the resulting neural networks are still expressive is very much in question.

The proposal is to first build a neural network g(x) that has a Lipschitz constant of lamda with respect to L1 norm, and then the overall model is
 f(x) = g(x) + lamda * sum_over_mono_set (x_i)
where x_i's are the subset of input features that we have the domain knowledge about monotonicity. This can easily be extended to when f(x) and g(x) are vectors, and we can have a different set of x_i's for each dimension in f(x), depending on the domain knowledge.

The main problem is in building g(x). For a linear layer, the L1 norm of the weight matrix is the max over sum of absolute values across each row. For g(x) which may have multiple layers, its Lipschitz bound from linear layers is the product of the said max-row-sum values across layers. This is very restrictive for larger networks and much more restrictive than past works that work with L2 Lipschitz bounds.
On pages 4-5, the authors argue that using GroupSort instead of ReLU would help. That is not true, and choosing non-linearities cannot solve the problem of restrictive linear layers.
If, for a particular application, it is sufficient to have g(x) as one linear layer plus some non-linearity, the proposed method should work fine. If two layers, this becomes questionable. For deeper networks, hidden dimensions need to be very small for this method to be viable. For general neural network architectures, I see little hope with the proposal.


**Summary Of The Paper:**

This paper proposes a method to build neural networks where an output is provably monotonic with respect to certain inputs. This is intended for applications where domain knowledge exists for such monotonic relations. For such applications, the benefits are better robustness and better interpretability.

**Summary Of The Review:**

This paper may get rejected for breaking blind review policy.

If not: this paper is working on a meaningful problem, however the proposed technique has significant limitations. It may work for very small neural networks but is not applicable to general architectures.

---

> ### Author Response · Authors · 2022-11-11
> **Response to tc6i Part I**
>
> Dear Reviewer tc6i -- Thank you for your detailed review.
>
> We will first answer the content section of the review. It is possible that the description of the method leaves some room to question the expressiveness of the proposed architecture. Because concerns about expressiveness can have a number of aspects, and for the sake of general clarity, we will attempt to cover every interpretation and kindly ask for clarification in case your concerns have not been addressed properly.
>
> We identify three important aspects:
>
> 1. The choice of $L^1$ over any other $p$-norm: Defining the target function class.
> 2. The architecture’s universal approximation of the target function class: $\mathrm{Lip}^1$ functions with respect to $L^1$ norm for row vector inputs (or equivalently, $L^{\infty}$ for column input vectors).
> 3. The expressiveness of the particular implementation proposed in modeling the target function class.
>
> We updated the submission (changes in blue) to add clarity to each of these points and summarize them here:
>
> 1. Choice of $L^1$:
> It is true that constraining a column vector $\textbf{v}$ to $||\textbf{v}||_1 \leq 1$ is more constraining than $||\textbf{v}||_2 \leq 1.$ Our goal is to constrain gradients of $g(\textbf{x})$ with respect to the inputs.
> However, we consider those gradients to be row vectors, as the batch dimension usually comes first. The $L^1$ norm for a row vector is equivalent to the $L^{\infty}$ norm for a column vector, which is the "least constraining" norm. Figure 1 intends to motivate our use of the $L^1$ norm and convey the idea of minimally constraining the gradient, such that the bound can be achieved in each feature individually. This is a requirement such that $f(\textbf{x})$ can model every monotonic Lipschitz function.
> For more intuition, imagine fitting the target function $h(x, y) = 0$ with this architecture. It is $\mathrm{Lip}^1$ and monotonic and should thus we should be able to approximate it with our parametrization $f(x, y)$.
> The skip connection gives a contribution $x+y$. Thus, $g(x, y)$ should fit $-x-y$, i.e. have gradient $(-1, -1)$. The only norm that permits $||\nabla g||_p \leq 1$ and $\nabla g = (-1,-1)$ is the $L^1$ norm (or $L^{\infty}$ for column gradients).
>
> 2. Universal approximation:
> Weight-normed architectures can encounter the problem in which stacking multiple constrained layers leads to weak performance because each layer may have small gradients in some region of the input resulting in the entire architecture being unable to model even certain simple functions (we added an example with a gradient maximizing function in the appendix). This is a well-studied problem in the literature and is commonly known as gradient norm attenuation. Combining GroupSort (or potentially other Gradient Norm Preserving functions such as the Householder activation) with the normalization scheme in Eq(11) yields a universal approximator of Lipschitz functions as proven in the references. That is, for a large enough network, this architecture is able to model any Lipschitz continuous function arbitrarily well. While no such proof exists for the normalization scheme in Eq(10), we find that both versions perform equally well in practice.
>
> 3. Expressiveness of the implementation:
>     For practical purposes, it is not sufficient to be a universal approximator. It is also important to be able to model an arbitrary function from the function class efficiently. We show empirical results comparing the performance of our architecture with prior work. Since we can retain the same level of performance while using a much smaller number of parameters, it is clear that our architecture is in fact more expressive than the state-of-the-art in these benchmarks. Outside of these benchmarks, to illustrate that we can indeed model complex nonlinear functions in practice, we have also included a specific example in Appendix A. We set up a regression problem to a target function that has $||\nabla||_1 = 1$ almost everywhere and show that we can attain good performance with a deeper and wider network. Note again that this function can be fit with neither $L^1$  nor $L^2$-constrained ReLU networks.
> Finally, we should mention the fact that, in our implementation of $\mathrm{Lip}^1$ networks, the constraints on each layer are independent of each other. Furthermore, **the constraints on the columns are also independent of each other** (see Eq(10)) since we simply require the abs column sum to be less than 1 for each individual column. This implementation does not get more restrictive with more layers and in fact, acts like an unconstrained network in certain regimes because no normalization is used when the $L^1$ norm of the column is already smaller than 1 (regardless of what other columns are doing).
>
> EDIT:
>
> Dear Reviewer tc6i, are there additional aspects that we should clarify before a final decision has to be made?

---

> > ### Author Response · Authors · 2022-11-11
> > **Response to tc6i Part II**
> >
> > In regards to the concerns about breaking the blind review policy:
> >
> > To the best of our understanding, we have not broken any of the blinding rules. We took several steps to anonymize our submission:
> > 1. We mention no names, have no acknowledgments, and redacted the GitHub link.
> > 2. We redacted a reference to prior work that could identify us.
> > 3. We copied the relevant parts of the repository into the supplemental materials such that we cannot be identified by looking through the requirements.yml and searching for the corresponding repository.
> > 4. We renamed the folder of the copied repository in the supplemental material such that the GitHub repository cannot be found by searching for the folder’s name in PyPI or Conda.
> >
> > Several quotes from the paper are listed in the review, but none of them imply any affiliation. The most that can be deduced is that we are familiar enough with the problem described in the paper to put forward a novel architecture to solve it. Indeed, each of the excerpts quoted highlights information that is available in the public domain, i.e. no particular membership or affiliation with any institution is needed to obtain this information. As for the data used (simulated samples), LHCb makes these available to non-collaboration members in situations like this where the goal of the work is not to make a physics measurement or projection, but instead to improve the operations of the experiment.
> >
> > The reason we mention the LHCb experiment at all is to draw attention to a realistic practical application of the algorithm in a pioneering and high-stakes environment. Even if LHCb affiliation was assumed, it is highly uninformative. LHCb is a collaboration with over 1000 members from 86 institutes from all over the world that regularly collaborates with researchers from both academia and industry (see publications).
> > Therefore, we argue that the blind review policy was not broken.

---

> > > ### Comment · Program_Chairs · 2022-11-28
> > > **msg from SPC**
> > >
> > > Reviewer - I don't see any clear violation of double blind policy. If you see it, please raise it to AC who can contact us with details.

---

> > ### Author Response · Authors · 2022-12-13
> > **Request for additional review**
> >
> > Dear Reviewer tc6i,
> > Please reconsider your initial assessment of the paper given the additional information provided in the updated version of the paper and the answer we gave above. We are confident that all concerns have been addressed.

---

> > > ### Comment · Reviewer_tc6i · 2022-12-14
> > > **thanks for the responses, however...**
> > >
> > > As said before, I leave the decision to the Area Chair on the blind review policy.
> > >
> > > On the universal approximation claim. According to Anil et al. 2019, the scheme by equation (11) can be a universal approximator with some additional conditions. The reason is that one can use the first linear layer (p,inf norm bounded) to represent all the piece-wise linear planes needed, -- each channel being one plane, -- and that subsequent layers (inf norm bounded) essentially do offsetting and taking max/min to put the planes together for a piece-wise linear approximation of the target function. This implies an arbitrarily wide first layer, and is only a theoretical existence for any moderately complex task. This has little to do with expressiveness on real tasks.
> > >
> > > On tasks with larger and deeper networks, the issue of restrictive Lipschitz bound will quickly become a problem, as I wrote in the original review. In fact, Anil et al. 2019 acknowledged so too: "While these constructions rely on constraining the inf-norm of the weights, constraining the 2-norm often makes the networks easier to train", where "these constructions" refers to the universal approximation proof for networks like (11). That's my point: theoretical existence of universal approximation does not equal expressiveness of these networks.
> > >
> > > Appendix A and B are 1) both toy examples and 2) both not monotonic! Appendix A is almost custom-designed for a network like (11), there are 64 piece-wise linear regions and one can hand-write a network like (11) with width 64 to represent it exactly. To claim expressiveness, let's start with MNIST.
> > >
> > > In summary, my assessment remains the same: whether the proposed neural networks are still expressive is very much in question.

---

### Decision · Program_Chairs · 2023-01-20

**Decision:**

Accept: poster

**Justification For Why Not Higher Score:**

Limited scalability

**Justification For Why Not Lower Score:**

Is good at what it does.

**Metareview: Summary, Strengths And Weaknesses:**

This paper proposes a guaranteed monotonic architecture by correcting for the Lipschitz constant of a network. The models are very small but perform well on standard datasets. While there is a clear limitation in the lack of scalability of this approach, there is also value in having strong models for smaller problems, and therefore this paper can be accepted.

The paper talks a lot about being state-of-the-art but is missing a discussion of and empirical comparison with Sivaraman, Aishwarya, et al. "Counterexample-guided learning of monotonic neural networks." Advances in Neural Information Processing Systems 33 (2020): 11936-11948. This is something to be corrected for the final version of the paper.

**Note From Pc:**

if the above contains the word "oral" or "spotlight" please see: "oral" presentation means -> notable-top-5% and "spotlight" means -> notable-top-25%. As stated in our emails, we are disassociating presentation type from AC recommendations